# Effect of Long-Term Aging on the Microstructural Evolution in a P91 Steel

**DOI:** 10.3390/ma15082847

**Published:** 2022-04-13

**Authors:** Hongchang Zhao, Zixi Wang, Xi Han, Mingjia Wang

**Affiliations:** 1Key Laboratory of Metastable Materials Science and Technology, College of Materials Science and Engineering, Yanshan University, Qinhuangdao 066004, China; hczhaoysu@163.com; 2CITIC Dicastal Co., Ltd., Qinhuangdao 066011, China; 3Department of Chemical Engineering and Modern Material, Shangluo University, Shangluo 726000, China; xihan202203@163.com

**Keywords:** aging, laves phase, coarsening kinetics, orientation relationship

## Abstract

The precipitation and growth mechanism of the Laves phase and the coarsening behaviors of Laves phase, M23C6 and MX carbonitrides have been emphatically investigated in P91 steel at 625 °C under different aging conditions. After long-term aging at 625 °C (>1500 h), it was observed that the Laves phase grew rapidly in the region near the M23C6 carbide once precipitated, and further gradually completed the engulfment process until the M23C6 carbide particles disappeared. Furthermore, a new crystallographic orientation relationship between M23C6 carbides and Laves phase has been observed at 625 °C for 5000 h, which is {0001}_Laves_∥{111}_M23C6_, <11$\bar{2}$1>_Laves_∥ <011>_M23C6_. The coarsening behaviors of Laves phase, M23C6 carbides and MX carbonitrides have been emphatically investigated, conforming to the existing ripening model of multicomponent alloys. The coarsening rates for the Laves phase, M23C6 and MX have values of ~32.2 (≥5000 h), 5.3 and 0.6 nm/h^1/^^3^, respectively.

## 1. Introduction

ASME P91 tempered martensitic ferritic steels are widely applied in the energy industry as critical structural components in ultra-supercritical thermal power units, which operate in the creep temperature range up to 866 K due to its excellent elevated temperature mechanical properties, creep performance and oxidation resistance [1,2]. After normalising and tempering treatments, these heat-resistant steels usually consist of a tempered martensitic matrix, MX carbonitrides and M23C6 carbides [3]. After long aging or creep process, large irregularly sized Laves phases form in the steel [4,5].

Cui et al. [6] investigated two micro-mechanisms of the nucleation and growth for the Laves phase appearing in steel X12. One is that the Laves phase particles precipitate preferentially on the prior austenitic grain boundaries and martensitic slat boundaries.

Another is that the Laves phase appears in the vicinity of the M23C6 particles and grows rapidly until it engulfs the M23C6 particles, which are consistent with other research results [7,8].

In 9–12% Cr ferritic heat-resistant steels, the size and stability of the subgrain has a crucial influence on creep strength [9,10,11]. While the type and size of the precipitated phase plays a predominant role in controlling the size of the subgrain, which is attributed to the fact that the size of the M23C6-type carbides is comparable to the thickness of the subgrain, with the fine M23C6 effectively hindering the migration of subgrain boundaries due to the Zener drag effect, and the nanoscale MX carbides improving creep strength by impeding dislocation movement [12].

There are two opposing views on the effect of Laves on the creep strength of 9–12% Cr steels. One is that the Laves phase contains a high content of W elements and therefore the diffusion of W elements from the matrix during the nucleation and growth of the Laves phase reduces its solid solution strengthening effect [13,14]. Secondly, it is believed that the fine Laves phase can be beneficial to creep strength by hindering subgrain boundary migration through the Zener drag effect [15], which is similar to the effect of M23C6.

Keisuke et al. [16] investigated the orientation relationship between Fe2W and α-Fe in Fe-10Cr-1.4W-4.5Co martensitic stainless steel as (201) Ferrite//(10$\bar{1}$4)Laves, <214> _Ferrite_//<1$\bar{2}$10>_Laves_. Abe F. [9] showed that the orientation relationship between Fe2W and α-Fe in 9Cr-4W heat-strength steel is (10$\bar{1}$0)_Laves_//(111)_Ferrite_, <0$\bar{1}10$>_Laves_∥<101>_Ferrite_. Xu [7] showed that Fe2W and α-Fe conform to the Burgers orientation relationship, i.e., (011)_Ferrite_//(0001)_Laves_, <11$\bar{2}$0>_Laves_∥<1$\bar{1}$1>_Ferrite_.

Therefore, this paper focuses on investigating the precipitation and growth mechanism of the Laves phase of P91 steel after aging at 625 °C under different aging conditions; the effect of nucleation sites on precipitation and growth behavior of Laves phase and the crystallographic orientation relationship between Laves phase and M23C6 carbides; and the coarsening kinetics of the precipitated phase after different aging conditions is also the focus of this paper.

## 2. Materials and Methods

Chemical composition of the steel used in this study is shown in Table 1. The elemental composition of the P91 steel sample was measured by direct reading spectrometer (SPECTROMAXx). The material was austenitized at 1050 °C for 1.5 h, followed by tempering at 760 °C for 4 h. Subsequently, long-term aging experiments were carried out at 625 °C for 750 h, 1500 h, 3000 h, 5000 h, 8000 h, 10,000 h and 15,000 h respectively.

The Z-contrast between the different types of precipitated phases and the matrix shows a significant difference in backscattered electrons (BSE) images, and a contrast greater than approximately 10% is required to clearly distinguish the different phases in a backscattered image [17]. The results of Reuter [18] show that Z-contrast between Mo-rich Laves phase precipitates and P91 matrix obtained by calculation is approximately 14%, thus Laves phase can be successfully recognized by SEM–BSE. and further analyzed by energy-dispersive spectrum (EDS) for its chemical composition.

The specimens after long-term aging for SEM-BSE were prepared by grinding and polishing, subsequently, the microstructure was then etched using a mixture of copper chloride, hydrochloric acid and alcohol (1:20:20) for approximately 10 s.

The microstructural evolution of various precipitates at different stages were analyzed by transmission electron microscope (Talos-F200X, FEI Corporation, Hillsboro, OR, USA), thin samples for TEM were first mechanically thinned to 30 μm, and then cut into 3 mm diameter discs. Finally, the double jet electro-polishing method was applied to obtain the thin foils using a 7% solution of perchloric acid in ethanol. The orientation relationships among the different precipitated phases can be revealed by means of electron backscatter diffraction (Hitachi, Japan).

The samples for the EBSD analysis were prepared by mechanical grinding and then polished with diamond paste and final further electropolishing at 5 C for 30 s with a mixture of 6% perchloric acid and 94% ethanol.

## 3. Results and Discussion

### 3.1. Precipitation Behavior for Laves Phase, M23C6 Carbides and MX Carbonitrides at Different Aging Times

In the original condition, Only M23C6 carbides and MX carbonitrides were seen in P91 steel except the Laves phase (Figure 1). The Laves phase was discovered at the prior austenite grain boundaries and martensite slat boundaries for 1500 h.

Figure 2a–d are SEM images and correspond to the BSE image of the microstructure aged at 625 °C for 3000 h and 5000 h. Figure 2a,b clearly illustrates the preferential nucleation sites for Laves phase are the prior austenite grain boundaries and martensite lath boundaries.

A single grain identified as the Laves phase nucleated next to an M23C6 grain (shown in Figure 2c,d) was mapped using XDS (as shown in Figure 2e). The results demonstrate that the grain is enriched with Mo, Fe and Si elements with respect to the matrix, which indicates it has an approximate Fe2Mo composition, while the segregation of Si element at carbides/ferrite interfaces can be attributed to the strong contribution of Si element to the formation of the Laves phase [19,20]. Meanwhile, the grain area is depleted with C elements. In addition, there is a clear tendency for carbon atoms to segregate around the phase interface between the Laves phase and matrix.

As shown in Figure 3a, the corresponding SAED patterns and chemical composition of the Laves phase and M23C6 aged at 625 °C for 10,000 h are investigated. Furthermore, it is noted that most of the Laves phase particles precipitate near the M23C6 carbide after 10,000 h and gradually engulf the nearby Cr-rich M23C6 carbide, and only a small portion of the M23C6 carbide do not undergo the above behavior, which is consistent with the results of Xu and Cui et al. [6,7]. Most of the Laves phase particles had almost completed the engulfment process, resulting in a rapid increase in size and very irregular shape, as shown in Figure 3b.

Xu et al. [6] investigated two micro mechanisms of the nucleation and growth for Laves phase in 10% Cr steel. One is that the Laves phase particles precipitate preferentially on the prior austenitic grain boundaries and martensitic slat boundaries and gradually grow by means of multicomponent diffusion at preferred nucleation sites [21,22], which is a relatively independent approach to nucleation.

Another is that the Laves phase appears in the vicinity of the M23C6 particles and grows rapidly until it engulfs the M23C6 particles, which is mainly attributed to the strong tendency of Si and P elements to segregate at the M23C6/ferrite interface, and greatly facilitating the formation of the Laves phase [19,20], which are in agreement with research result of previous studies [7,8].

### 3.2. The Coarsening Behaviors of Laves Phase, M23C6 and MX

Figure 4 shows the evolution of the average diameter, the number density and area fractions of Laves phase, M23C6 and MX particles at 625 °C with different aging times. Statistics for all the particles of the precipitated phases were obtained from a number of TEM images, and at least 100 particles of each precipitate were counted at each given aging condition by Image Pro Plus software.

The average diameter statistics for the three precipitated phases from Figure 4a show that the average diameter of M23C6 first increases rapidly (<1500 h) and then the growth rate of the particles becomes slower as the aging time increases from 1500 h to 15,000 h. In contrast, the Laves phase particles show a more rapid increase in size compared to M23C6 once generated, which start from 155 nm to 537 nm with the aging time extended to 15,000 h.

From the above statistics of the mean size of the Laves phase and M23C6 particles, it appears that the average size of both increases steadily from the original state to age for 15,000 h, with no tendency to enter a stabilization period, which means that the average size will increase further as time continues to increase. Compared to the two precipitated phases mentioned above, the MX carbide size is rather stable, which maintains a slight increase from 52 nm to 65 nm after aging for up to 15,000 h.

Figure 4b illustrates the coarsening behavior of the Laves phase, M23C6 and MX. The evolution of the mean particle size d as a function of t^1/3^ for M23C6 and MX is in accordance with a linear relationship, and the coarsening kinetics of multi-component alloys can be formulated by Equation (1) [23], which is consistent with the Lifshitz-Slyozov-Wagner (LSW) model of Ostwald ripening during long-term aging [10,24]. Nevertheless, the increase in particle size satisfies the linear relationship very well only from 5000 h to 15,000 h.
(1)$$\bar{r}=kt^{1/3}$$
where *k* is the coarsening rate, *t* is the thermal aging time and $\overset{¯}{r}$ is the average radius of precipitate particles.

The coarsening rate *k* is obtained from the slope of the linear fit curve in Figure 4b, giving values of ~32.2 (5000 h), 5.3, and 0.6 nm/h^1/3^ for Laves phase, M23C6 and MX, respectively. The evolution of the number density of M23C6, Laves phase and MX with different aging times is illustrated in Figure 4c.

Figure 4c illustrates that there is a significant decrease in the number density of M23C6 from its original state to 15,000 h, and the increase in the number density of MX is rather slow from 0.14 to 0.21 particles mm^−2^. In contrast, the number density of the Laves phase maintains a small increase from 0.017 to 0.135 particles mm^−2^. Statistical analysis of the number density illustrates that the number density values of M23C6 still remain much greater than those of the Laves phase and MX for 15,000 h.

Area fractions for each type of precipitate was calculated by dividing the total of projected areas of the particles by the area of the image used for statistics. The volume fractions of M23C6 and MX remain essentially stable aged for up to 15,000 h in value over the long aging process. Nevertheless, that of the Laves phase gradually increase from 0.25% after 1500 h to 1.96% after 15,000 h, as shown in Figure 4d.

Statistical analysis of average size, number density and area fraction showed that the MX carbides were fairly stable, the growth of the M23C6 carbide particles is in accordance with the Ostwald ripening model and the Laves phase particles did not reach thermodynamic equilibrium, which is in accordance with previous studies [25,26].

### 3.3. Orientation Relationships between Laves Phase and M_23_C_6_

Figure 5 shows the EBSD profile of P91 steel with CI >0.2 obtained by aging for 5000 h. The black lines on the grain boundaries indicate a crystal orientation difference of 15°–180° and the white lines indicate a crystal orientation difference of 2°–15°. Figure 5a shows the crystal orientation distribution and Figure 5b shows the phase distribution corresponding to Figure 5a. Three phases are present in the figure, the red one is the matrix ferrite, the green one is M23C6 and the yellow one is the Laves phase.

Figure 5c,d shows the PF diagrams of regions C and D in Figure 5a respectively, from which it can be seen that both the M23C6-type carbides precipitate on the grain boundaries and within the crystal have a certain orientation relationship with the ferrite matrix, i.e., {011}Ferrite∥{111}M23C6, <111> Ferrite∥ <011> M23C6, and Figure 5d illustrates that the M23C6 carbide has a certain orientation relationship with the Laves phase, i.e., (0001) Laves∥(111) M23C6, <11$\bar{2}$0> Laves∥ <011> M23C6. The dense row surface of the Laves phase is not parallel to the dense row surface of the ferrite matrix.

Keisuke Y et al. [16] investigated the orientation relationship between Fe2W and α-Fe in Fe-10Cr-1.4W-4.5Co martensitic stainless steel as (201)_Ferrite_∥(10$\bar{1}$4)_Laves_, <214>_Ferrite_∥ <1$\bar{2}$10>_Laves_. Abe F et al. [9] showed that the orientation relationship between Fe2W and α-Fe in 9Cr-4W heat-strength steel is (10$\bar{1}$0)_Laves_∥(111)_Ferrite_, <0$\bar{1}10$>_Laves_∥<101>_Ferrite_. Xu [7] showed that Fe2W and α-Fe conform to the Burgers orientation relationship, i.e., (011)_Ferrite_∥(0001)_Laves_, <11$\bar{2}$0>_Laves_∥<1$\bar{1}$1>_Ferrite_. The experimental results in this paper show that the relationship between the Laves phase and the substrate in P91 does not conform to any of the three positional relationships above.

## 4. Conclusions

The microstructural evolution of different phases in P91 steel after long-term aging at 625 °C for up to 15,000 h have been studied. The main results can be concluded as follows:
1Laves phase, after long-term aging (>1500 h) at 625 °C, grew rapidly once precipitated. In addition, some Laves phases are formed in the regions adjacent to M23C6 particles.2A new crystallographic orientation relationship between M23C6 carbides and Laves phase has been observed at 625 °C for 5000 h, the new orientation relationship is{0001}_Laves_∥{111}_M23C6_, <11$\bar{2}$1>_Laves_∥ <011>_M23C6_.3The coarsening behavior of the Laves phase is mainly affected by the phagocytic growth mechanism before 5000 h. The large irregular Laves phase precipitated at grain boundary or phase boundary will become an effective crack source, which will reduce rupture life of heat-resistant steel.4The coarsening behaviors of Laves phase (more than 5000 h), M23C6 carbides and MX carbonitrides have been investigated, which confirms the existing ripening model of multicomponent alloys. The coarsening rates for the Laves phase, M23C6 and MX have values of about 32.2 (≥5000 h), 5.3 and 0.6 nm/h^1/3^, respectively.

## Figures and Tables

**Figure 1 materials-15-02847-f001:**
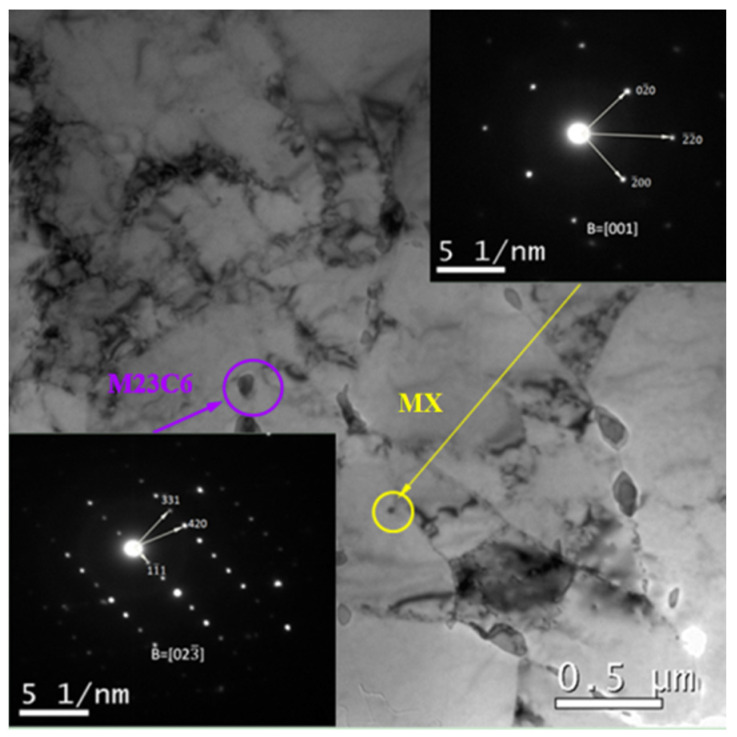
TEM BF image and SAED patterns of the P91 steel for initial state.

**Figure 2 materials-15-02847-f002:**
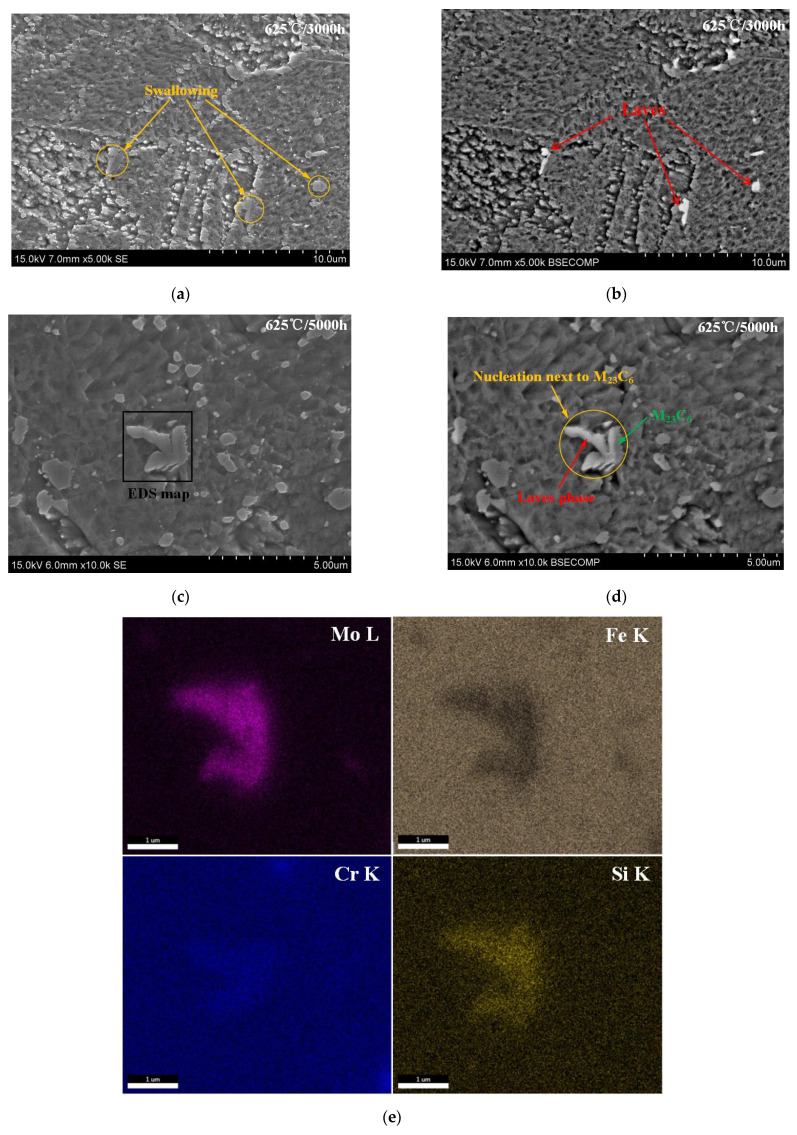
SEM, BSE and EDS images of the P91 steel aged at 625 °C: (**a**) SEM image for 3000 h, (**b**) the corresponding BSE image for (**a**) SEM image, (**c**) SEM image for 5000 h, (**d**) the corresponding BSE image for (**c**) SEM image, © EDS X-ray maps of selected area in (**e**) for Mo, Fe, Cr, Si, (**f**) EDS X-ray maps of selected area in (**c**) for C.

**Figure 3 materials-15-02847-f003:**
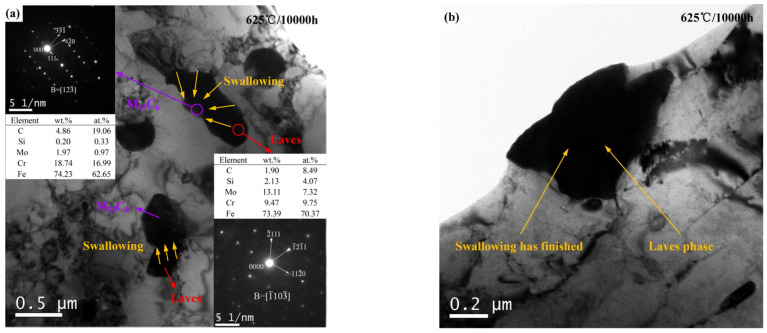
The growth mechanism for Laves phase aged at 625 °C for 10,000 h: (**a**) TEM BF image including SAED patterns and chemical composition of particles, (**b**) TEM image, the swallowing process has completed.

**Figure 4 materials-15-02847-f004:**
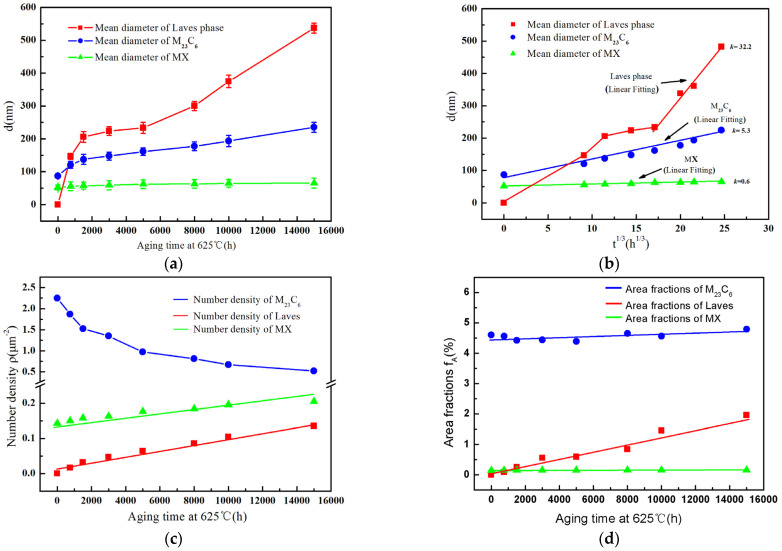
Coarsening kinetics, number densities and area fractions of precipitates the P91 steel during long-term aging: (**a**) d as a function of t for M23C6, Laves phase and MX; (**b**) d as a function of t^1/3^; (**c**) Number densities; (**d**) Area fractions.

**Figure 5 materials-15-02847-f005:**
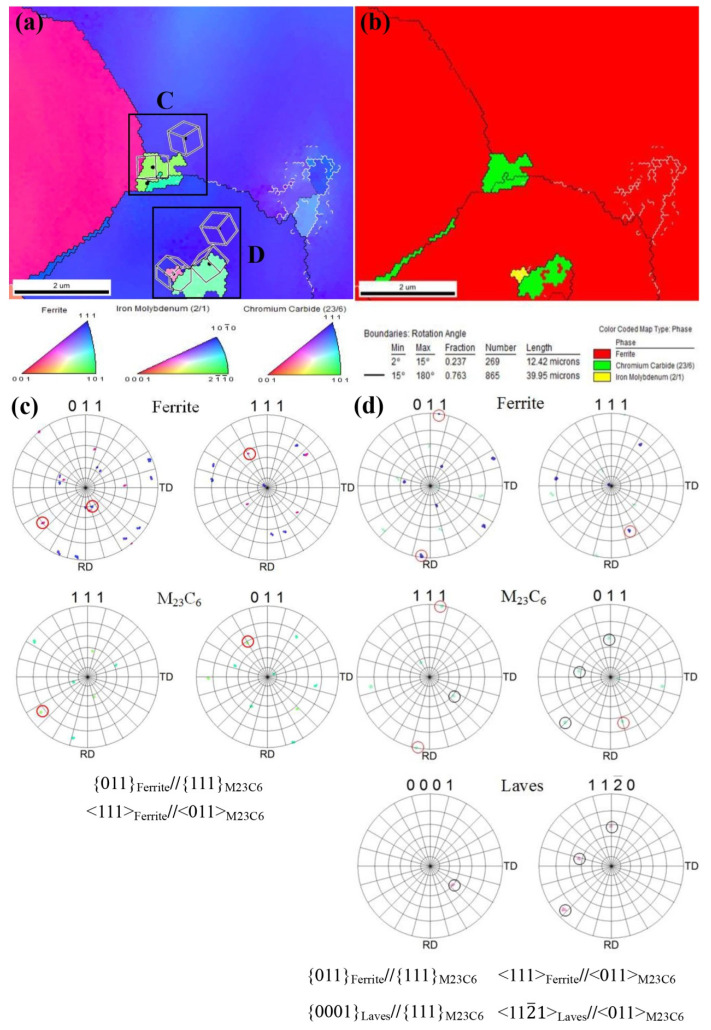
EBSD maps of the P91 steel aged at 625 °C for 5000 h: (**a**) image quality (IQ) and inverseflame pole figure (IPF), (**b**) Phase distribution, (**c**) Pole Figure (PF) of M23C6 and Ferrite in region C cropped from (**a**), (**d**) PF of M23C6, Laves phase and Ferrite in region D cropped from (**a**).

**Table 1 materials-15-02847-t001:** The chemical composition (wt.%) of the P91 steel.

Element	C	Si	Mn	P	S	Cr	Ni	Mo	V	Nb	N	Fe
wt.%	0.10	0.30	0.5	0.017	0.003	9.0	0.7	1.0	0.20	0.06	0.04	Bal.

## Data Availability

The datasets generated during and/or analyzed during the current study are available from the corresponding author on reasonable request.

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
