# Peer review of "Effect of Long-Term Aging on the Microstructural Evolution in a P91 Steel"

_materials, 2022, doi:10.3390/ma15082847_

Round 1

Reviewer 1 Report

The manuscript "Effect of long-term aging on the microstructural evolution in a P91 steel" by Hongchang Zhao et al. reports on the electron microscopy study on the microstructural evolution of P91 steel upon aging at 625oC for many hours. The authors mainly focus on the behavior of three precipitate phases, viz., intermetallic Laves phase, M23C6 carbide and MX carbonitride phases. The topic is of general interest to materials scientists and well appropriate for the scope covered by Materials.

There are several issues that should be addressed by the authors before a positive decision on the manuscript acceptance could be made. They are detailed below.

1) The origin of elemental composition of the P91 steel sample reported  in Table 1 should be indicated. Was it determined experimentally (which technique?) or is is just a standard composition anticipated for this steel grade?

2) Essential experimental parameters used for SEM-BSE,  EDS, TEM, EBSD should be given.

3) The procedure applied to recognize and identify individual grains to collect statistics given in Figure 4 should be described in more detail. Probably, a sample full-area image with marked grains would be informative.

4) Idealized (approximate) chemical compositions and crystallographic references should be given for the  three focal phases, including Laves, M23C6, and MX.

5) Elements should be labelled  as C, Si, Mo, Cr, Fe, and not C K, Si K, Mo L, Cr K, Fe K in insets to Figure 3 (elements rather than X-ray fluorescence lines are meant there).

6) The light green marks related to M23C6 grains are poorly seen in Figures 1, 2, and especially 3. A different color code is recommended.

7) EDS mapping results should be disccussed in more detail. For instance, the phrase

The EDS X-ray image of Figure 2c shows that in Figure 2e-f, the Laves phase can be found to be composed of Mo, Fe, Cr and Si

should rather read

A single grain identified as the Laves phase nucleated next to an M23C6 grain (shown in panels (c) and (d) of Fig. 2) was mapped using XDS (as shown in panel (e) of Fig. 2). The results demonstrate that the grain is enriched with ...elements with respect to the matrix, which inidcates it has an approximate ... composition. Meanwhile, the grain area is depleted with ... elements. Carbon atoms panel (f) of Fig. 2) tend to segregate at the grain boundary of the Laves grain...etc.

8) Figure 4 has currently no panel labels (a)-(d). The legend given in panel (c) is incorrect. It should read Number density rather than Area fractions. 

9) The equation (1) with the cube-root dependence of mean grain size on time represents the so called Lifshitz-Slyozov-Wagner (LSW) model of Ostwald ripening. This should be explicitly outlined by the authors with appropriate referencing.

10) The acronyms PF (pole figure, not polar!) and IQ (image quality?) should be decoded in the text .

11) Finally, the authors should put their results in a broader scientific context. in Conclusions. What is the main value of their findings? Is it good or bad for mechanical toughness of high-T steel parts that Laves phase grains monotonically grow without an asymptotic saturation? Should this be promoted or rather avoided in real-life applications?

Author Response

Dear Reviewers,

    A point-by-point response has been made to your review comments, please refer to the attached document for the response.

Yours sincerely,

Dr. Zhao

Reviewer 2 Report

  • Abstract is too general. Kindly revise the abstract.
  • Introduction is too limited. Detail literature review is essential.
  • Conclusion is too weared and confusing.
  • English should be improved.

Author Response

(The authors gave the same response as above.)

Reviewer 3 Report

Very interesting work, considering the 15.000 h heat treatment duration (great effort and patience) and modelling the precipitate growth. The analysis and the analytical tests chosen are all solid and reliable which makes the conclusion to be justified. I almost found no criticism or comments.

My only question to the authors is about the starting material. I think a bit more detailed information may be required about the raw material.

What was the production method? Thickness? Treatments and history of the material?

Author Response

Dear Reviewer,

A point-by-point response has been made to your review comments, please refer to the attached document for the response.

Yours sincerely,

Dr. Zhao

Round 2

Reviewer 1 Report

The authors have appropriately addressed my concerns related to the first version of the manuscript and thus I recommend acceptance of the revised manuscript in its present form.

Reviewer 2 Report

The latest version is adequate for publication